# Normative serum cortisol levels in second and third trimesters and their associated factors: A prospective cohort study from Sri Lanka

Ishani Menike[1ʘ], Shashanka Rajapakse[2,3ʘ], Gayani Amarasinghe[4], Janith Warnasekara[4], Nuwan Darshana Wickramasinghe[4], Thilini Agampodi[4,5], Suneth Buddhika Agampodi[4,5,6]*

1 District General Hospital Nuwaraeliya, Nuwara Eliya, Sri Lanka, 2 Graduate School of Cancer Science and Policy, National Cancer Center, Goyang-si, Gyeonggi-do, Republic of Korea, 3 Department of Physiology, Faculty of Medicine and Allied Sciences, Saliyapura, Anuradhapura, Sri Lanka, 4 Department of Community Medicine, Faculty of Medicine and Allied Sciences, Rajarata University of Sri Lanka, Saliyapura, Sri Lanka, 5 International Vaccine Institute, SNU Research Park, Seoul, Republic of Korea, 6 Section of Infectious Diseases, Department of Internal Medicine, School of Medicine, Yale University, New Haven, Connecticut, United States of America

ʘ These authors contributed equally to this work.
* suneth.agampodi@yale.edu

## Abstract

### Objective

To describe the normative serum cortisol levels during 25–29 weeks of POG and the association of maternal, psychological, and social factors on serum cortisol in the second and third trimesters in a cohort of pregnant women.

### Methods

All eligible pregnant women registered in the maternal care program in Anuradhapura district, Sri Lanka, from July to September 2019 were invited to the Rajarata Pregnancy Cohort (RaPCo). An interviewer-administered questionnaire-based symptom analysis and clinical assessment were conducted at baseline in the first trimester and at follow-up from 25 to 29 weeks POG. We assessed fasting early morning serum cortisol levels at the follow-up visit.

### Results

The study sample included 1010 pregnant women with a mean age in years and POG in weeks at baseline of 28 (±6) and 10 (±3), respectively. The mean (SD, 97% percentile) serum cortisol level in all pregnant women was 10.93 (±3.83, 20.95) μg/dL, with no significant difference between singleton and twin pregnancies ($p = 0.138$). None of the study participants had a cortisol level exceeding the upper limit of 42 μg/dL, and 464 (45.9%) had levels less than 10 μg/dL. Serum cortisol levels were higher

**Data availability statement:** All deidentified data of the participants included in the current study of the Rajarata Pregnancy Cohort (RaPCO) is deposited under the doi: https://doi.org/10.5281/zenodo.15074568 at the https://zenodo.org/records/15074568.

**Funding:** The original cohort study was supported by the Accelerating Higher Education Expansion and Development (AHEAD) Operation of the Ministry of Higher Education, Sri Lanka funded by the World Bank [grant number DOR STEM HEMS [6026-LK/8743-LK]]. The funding agency has no role in the design of the study and collection, analysis, and interpretation of data and in writing the manuscript.

**Competing interests:** The authors have declared that no competing interests exist.

in women with an advanced POG, with a mean of 10.33 µg/dL (95%CI: 9.68–10.98) at 24 weeks POG and 12.23 µg/dL (95%CI: 11.15–13.32) at 29 weeks POG ($p=0.049$). Primi-gravidity ($p=0.004$), history of miscarriage ($p=0.010$), BMI categories ($p=0.044$), and POG ($p=0.002$) were independently associated with serum cortisol levels in robust regression. An EPDS score of more than 9 was not associated with serum cortisol ($p=0.633$).

## Conclusion

The pregnant women in rural Sri Lanka reported a low mean serum cortisol level, which gradually increased with the POG. Significantly higher mean serum cortisol was associated with primi-gravidity, history of miscarriage, pre-pregnancy BMI, and POG at cortisol test, but not psychological factors.

## Introduction

Maternal stress during pregnancy plays a critical role in maternal and fetal well-being. Stress during preconception and the antenatal period adversely affects maternal and fetal health through the dysregulation of the hypothalamic-pituitary axis [1]. Maternal stress increases the risk of maternal complications, including eclampsia, premature labor, and postpartum hemorrhage [2–4]. Recent evidence synthesis indicates that maternal stress is implicated in intrauterine growth restriction, prematurity, and low birth weight [5]. It is also associated with developmental disorders and childhood physical and psychological disorders [6,7]. Thus, maternal stress during pregnancy and its causes should be explored in depth to ensure safe motherhood and the well-being of their offspring.

Currently, maternal stress is measured in two broad approaches: cortisol level and psychometric tools [8]. Different measures of cortisol, such as serum, urinary, hair, and diurnal cortisol, are currently used to assess stress [9]. Cortisol, the stress hormone, is considered a better objective indicator of physiological stress compared to the subjective evaluations conducted using psychometric tools [10–12]. Maternal cortisol, the final product of the hypothalamic-pituitary axis, is a key hormone in fetal development and maturation [13]. A recent study indicates that maternal serum total cortisol increased on average during pregnancy from 390 (±22) nmol/L in the 5th week to 589 (±15) nmol/L in the 20th week [14]. Current evidence suggests that high serum cortisol of more than 17.66 µg/dL (488 nmol/L) is significantly associated with developing postpartum depression [15]. However, studies from low socio-economic countries scarcely include measurement of cortisol levels in pregnancy [9].

The factors contributing to maternal stress are diverse, and perceived maternal stress is associated with circumstantial factors such as marital status, unplanned pregnancy, societal factors such as support from extended family, factors such as teenage pregnancy, domestic abuse, psychological factors, and economic factors [16]. Furthermore, emotional lability and physiological changes in pregnancy, such as weight gain, nausea, and insomnia, also contribute to significant stress during

pregnancy [17–19]. Elevated cortisol levels are known to be associated with maternal depression and adverse birth outcomes [20]. This is further compounded in complicated pregnancies [21].

Although Sri Lanka is a lower-middle-income country with comparatively high maternal health indicators, current evidence points to high levels of antenatal anxiety, depression, and maternal suicides [22–24]. There are no studies reporting serum cortisol levels or predictors of cortisol in pregnant women of Sri Lanka. The current study investigates the normative data of serum cortisol levels in the second and third trimesters in a large population-based cohort and explores the association of maternal biological and psychosocial stressors with the serum cortisol level in the second trimester.

## Methodology

### Study design

We conducted a population-based prospective cohort study. The detailed methodology has been published [25].

### Study setting

This study was conducted in Anuradhapura, geographically the largest district in Sri Lanka. It is predominantly rural with mostly Sinhalese (91%), followed by Sri Lankan Moors (8.2%), Tamils (0.6%), and other minority groups (0.2%) [26]. The female literacy rate is 91.7% and 37.1% of female adults participate in economic activities in Anuradhapura, Sri Lanka [27]. Each year, over 15,000 expectant mothers enroll in public antenatal care services in Sri Lanka. These services are delivered through divisional units within the Medical Officer of Health (MOH) areas. The district consists of 22 MOH areas, where all pregnant women are systematically registered by public health midwives (PHM), either during field visits or at their initial antenatal appointment.

### Study population

The target population comprised pregnant women residing in the Anuradhapura district, North Central Province, Sri Lanka, who were eligible for enrolment based on the following criteria: (i) registration with the national maternal care program between 1 July and 30 September 2019, (ii) confirmed pregnancy with a period of amenorrhea less than 12 weeks at the time of registration, and (iii) permanent residence within the study district. The national maternal care program achieves over 95% population coverage, enabling near-complete enrolment of eligible pregnancies into antenatal care services [28].

### Study sample

A total of 3,374 pregnant women were recruited for the original cohort study [29]. Of these, 1,450 participants attended the first follow-up visit, which occurred during the late second or early third trimester. Among those who attended the follow-up, 1,290 (89.0%) underwent serum cortisol testing. At this follow-up, gestational age was re-estimated based on first-trimester dating ultrasound scans. For the present analysis, a subset of 1,010 women with recalculated gestational ages between 25 and 29 weeks at the time of cortisol sampling was selected.

### Procedure

At the baseline, all participants underwent a detailed clinical interview, anthropometric assessments, and venipuncture for blood and serum investigations in addition to routine maternal care. Interviewer-based questionnaires were completed by medically trained data collectors to gather participants' sociodemographic, medical, psychological, and social information. Anthropometric measurements included weight, height, waist circumference, and hip circumference to assess body mass index (BMI) and waist-to-hip ratio according to standard protocols.

During the follow-up visit between 25 and 29 weeks of POG, fasting early morning serum cortisol levels were obtained by a trained phlebotomist after adequate resting for each participant, obtaining blood in a single prick. The analytical work was conducted by an accredited external laboratory using a fully automated analyzer based on the electrochemilumi-nescence immune assay method. The validated and culturally-adopted Edinburgh Postnatal Depression Scale (EPDS) was used to assess antenatal depression in the first (POG less than 13 weeks) and second visits (POG 25–29) [30,31]. Furthermore, attempts or ideation of deliberate self-harm and/or suicide, and intimate partner violence and/or abuse were assessed in the second trimester.

### Statistical analysis

The distribution of cortisol levels from 25 to 29 weeks of POG was assessed for the whole study sample and singleton pregnancies. We considered a serum cortisol level of 10 µg/dL to 42 µg/dL as normal [32]. The serum cortisol level was described and compared based on the trimester, number of fetuses, gravidity, parity, maternal age, contraceptive failure, history of miscarriage, BMI category, waist-to-hip ratio, glycemic status, demographic factors, including maternal and paternal ethnicity, religion, and highest grade completed in school, and factors on mental health. The mean compari-son of two groups and more than two groups was conducted with an independent sample T-test and one-way ANOVA, respectively, with a significance ($p$) of 0.05. One-way ANOVA was conducted to compare the mean serum cortisol levels of women with a POG between 24 and 29 weeks with Levene's test to test the heterogeneity of variances, and, if hetero-geneous, Games-Howell post hoc comparisons were used. The association between total EPDS score and components (anhedonia, anxiety, and depression) in the first trimester and the mean serum cortisol level was explored with an inde-pendent sample T-test or one-way ANOVA [33]. Variables found to be significant ($p < 0.05$) in the bivariate analysis were further evaluated using multivariable regression to identify independent associations with serum cortisol levels. In addition to these variables, marital status and maternal education were included in the model due to their potential confounding effects. Parity was excluded because of multicollinearity with gravidity. Given that serum cortisol was not normally distrib-uted, the dependent variable was log-transformed to satisfy the assumptions of linear regression and improve model per-formance. After comparing model diagnostics, robust linear regression was selected over standard ordinary least-squares regression, as it provides more reliable estimates in the presence of outliers and non-constant variance in residuals.

### Ethical considerations

Ethical approval for the study was obtained from the Ethics Review Committee of the FMAS, RUSL (ERC/2019/07). Informed written consent was obtained from the participant, and if the participant was less than 18 years of age, consent from parent(s) and/or guardian(s) and written assent from the participant were obtained before recruiting to the study.

## Results

### Characteristics of the study sample

The study sample included 1010 pregnant women between 15–45 years of age (Table 1). The mean (SD) age at conception was 27.6 (±5.7). There were 82 (8.1%) teenage pregnancies (age at conception less than 20 years), while most women belonged to the 25–29 age category (n = 351, 34.8%). In this study sample, 23 (2.3%) were not married. Most women were in their first (n = 317, 31.4%) or second pregnancy (n = 316, 31.3%). Contraceptive failure was the reason for pregnancy in 26 (2.6%) of pregnancies. A history of miscarriage was reported by 189 (18.7%) women. Most participants belonged to the normal body mass index (n = 361, 35.7%), and the waist-to-hip ratio of more than 0.85 (n = 349, 34.6%) categories.

### Normative serum cortisol data in 24–29 weeks of POG

The distribution of cortisol levels in this study was skewed right (Shapiro-Wilk = 0.86, p < 0.01). The mean (SD, 97.5% per-centile) serum cortisol level of the study sample was 10.93 (±3.83, 20.95) µg/dL, and there was no significant difference

**Table 1. Biological factors affecting cortisol levels in the second and third trimesters of pregnant women from Anuradhapura, Sri Lanka (N = 1010).**

| | N | Percentage (%) | Cortisol level (µg/dL) | | | Significance |
|---|---|---|---|---|---|---|
| | | | Mean | Standard deviation | Median | |
| **Age at conception (years)** | | | | | | F(5,1010)=4.54 |
| Less than 20 | 82 | 8.1 | 12.45 | 4.73 | 11.15 | *p* < **0.001** |
| 20-24 | 209 | 20.7 | 11.16 | 3.63 | 10.46 | |
| 25-29 | 351 | 34.8 | 10.93 | 3.65 | 10.26 | |
| 30-34 | 236 | 23.4 | 10.72 | 4.15 | 9.98 | |
| 35-39 | 115 | 11.4 | 10.01 | 3.11 | 9.90 | |
| More than 40 | 17 | 1.7 | 9.87 | 2.21 | 10.16 | |
| **Pregnancy** | | | | | | T=−1.48 |
| Singleton | 1005 | 99.5 | 10.92 | 3.81 | 10.30 | *p*=0.138 |
| Twin | 5 | 0.5 | 13.46 | 6.93 | 10.68 | |
| **Period of gestation at cortisol test (weeks)** | | | | | | F(5,1010)=3.71 |
| 24 | 75 | 7.4 | 10.33 | 2.82 | 10.28 | *p*=**0.002** |
| 25 | 182 | 18.0 | 10.55 | 3.38 | 10.18 | |
| 26 | 248 | 24.6 | 10.68 | 3.45 | 10.21 | |
| 27 | 254 | 25.1 | 10.84 | 3.95 | 10.10 | |
| 28 | 156 | 15.4 | 11.41 | 3.84 | 10.34 | |
| 29 | 95 | 9.4 | 12.23 | 5.33 | 11.15 | |
| **Body Mass Index** | | | | | | F(3,974)=3.30 |
| Underweight | 172 | 17.0 | 11.69 | 4.34 | 10.85 | *p*=**0.020** |
| Normal | 361 | 35.7 | 10.77 | 3.61 | 10.27 | |
| Overweight | 155 | 15.3 | 11.13 | 4.04 | 10.41 | |
| Obese | 286 | 28.3 | 10.60 | 3.65 | 9.96 | |
| **Waist-to-hip ratio** | | | | | | F(2,970)=1.64 |
| Less than 0.8 | 319 | 31.6 | 11.15 | 4.11 | 10.41 | *p*=0.195 |
| 0.8 to 0.849 | 302 | 29.9 | 11.09 | 3.94 | 10.42 | |
| More than 0.85 | 349 | 34.6 | 10.66 | 3.44 | 9.99 | |
| **Glycemic status** | | | | | | F(2,947)=1.47 |
| Gestational DM[a] | 54 | 5.3 | 10.39 | 3.37 | 9.46 | *p*=0.229 |
| Preexisting DM[a] | 17 | 1.7 | 9.74 | 1.97 | 9.86 | |
| Normoglycemia | 876 | 86.7 | 10.99 | 3.88 | 10.32 | |
| **Gravidity** | | | | | | F(3,1009)=5.51 |
| Primi | 317 | 31.4 | 11.62 | 4.35 | 10.59 | *p* < **0.001** |
| Second | 316 | 31.3 | 10.80 | 3.25 | 10.47 | |
| Third | 251 | 24.9 | 10.45 | 3.60 | 9.84 | |
| Four or more | 125 | 12.4 | 10.48 | 4.01 | 10.09 | |
| **Parity** | | | | | | F(3,1009)=7.70 |
| Nulliparous | 359 | 35.5 | 11.66 | 4.36 | 10.59 | *p* < **0.001** |
| One | 348 | 34.5 | 10.75 | 3.05 | 10.43 | |
| Two | 258 | 25.5 | 10.27 | 3.72 | 9.84 | |
| Three or more | 44 | 4.4 | 10.30 | 4.40 | 9.02 | |
| **History of miscarriage** | | | | | | T=−2.38 |
| Yes | 189 | 18.7 | 11.13 | 4.20 | 10.38 | *p*=**0.018** |
| No | 502 | 49.7 | 10.42 | 3.22 | 10.06 | |
| **Contraceptive failure** | | | | | | T=−1.51 |
| Yes | 26 | 2.6 | 12.05 | 5.66 | 10.61 | *p*=0.13 |
| No | 984 | 97.4 | 10.90 | 3.78 | 10.27 | |

[a]DM = diabetes mellitus.

(p=0.138) between the mean serum cortisol levels of singleton (10.92±3.81 µg/dL) and twin pregnancies (13.46±6.93 µg/dL) (Fig 1). None of the study participants had a cortisol level exceeding the upper limit of 42 µg/dL, and 464 (45.9%) had levels less than 10 µg/dL, with 236 (50.9%) in the second trimester and 228 (49.1%) in the third trimester. Serum cortisol levels gradually increased with the POG, with a mean of 10.33 µg/dL (95%CI: 9.68–10.98) at 24 weeks POG and 12.23 µg/dL (95%CI: 11.15–13.32) at 29 weeks POG (Fig 2). One-way ANOVA was conducted for those who are between 24 and 29 weeks, and it indicated a significant effect of POG on serum cortisol levels (F (5, 1010) =3.71, p=0.002) but with a small effect size ($\eta^2$=0.02). Levene's test was significant (p=0.002), indicating heterogeneity of variances, and, thus, Games-Howell post hoc comparisons were used. A significant difference was observed only between 24 weeks POG and 29 weeks POG (p=0.049), with no other comparisons reaching statistical significance.

## Association of serum cortisol levels with biological and maternal factors

A significantly higher mean serum cortisol level was detected in primi-gravida (one-way ANOVA F (3, 1009) =5.51, p<0.001), nulliparous women (one-way ANOVA F (3, 1009) =7.70, p<0.001), teenage pregnancies (T=3.78 p<0.001) and women with a history of miscarriage (T=−2.38, p=0.018) compared to their respective counterparts (Fig 3). There was a significant difference between the mean serum cortisol levels of underweight, normal weight, overweight, and obese individuals (one-way ANOVA F (3, 974) =3.30, p=0.020). However, there was no significant difference in the mean serum cortisol levels between the groups based on the waist-to-hip ratio (one-way ANOVA F (2, 970) =1.64, p=0.195) or the glycemic status (gestational diabetes mellitus, pre-existing diabetes mellitus, and normoglycemia) (one-way ANOVA F (2, 947) =1.47, p=0.229). Maternal ethnicity (one-way ANOVA F (2, 1000) =0.414, p=0.661), religion (one-way ANOVA F (2, 1010) =0.827, p=0.438), highest grade completed in school (one-way ANOVA F (2, 1003) =0.073, p=0.930) or paternal ethnicity (one-way ANOVA F (2, 1007) =0.300, p=0.741), religion (one-way ANOVA F (2, 1008) =0.614, p=0.541), highest grade completed in school (one-way ANOVA F (2, 998) =1.057, p=0.350) were associated with a higher mean serum cortisol level.

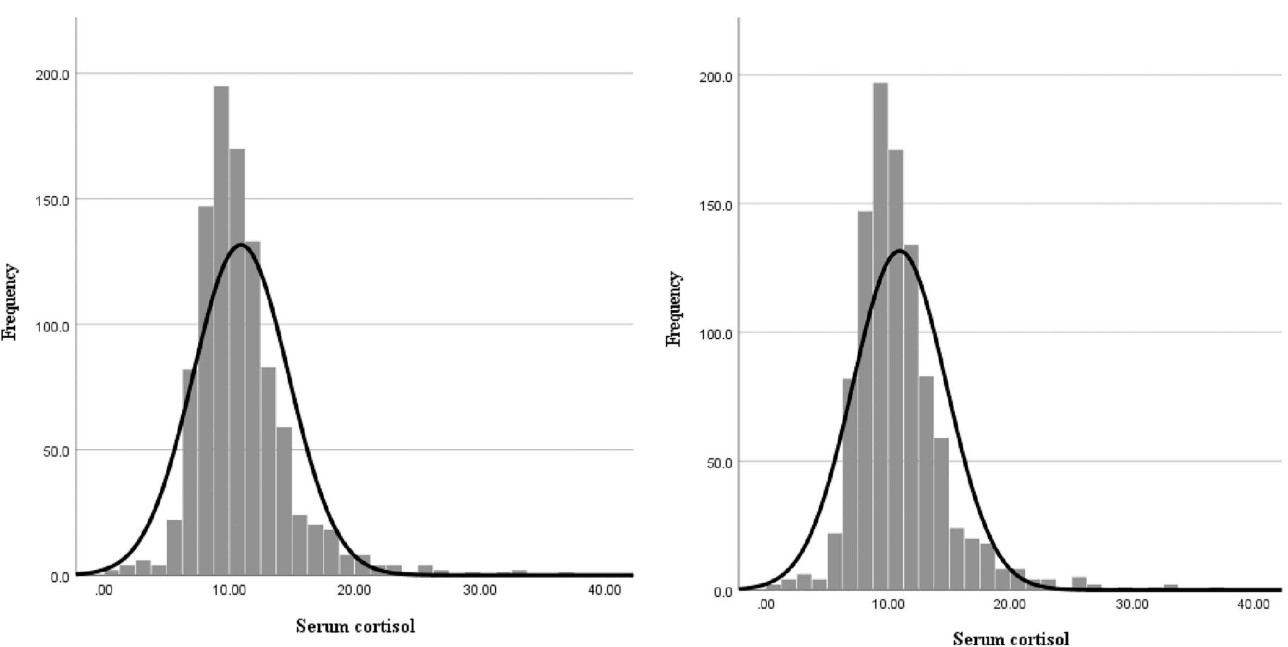

**Fig 1. Distribution of cortisol in all pregnant women (left) and in singleton pregnancies (right).**

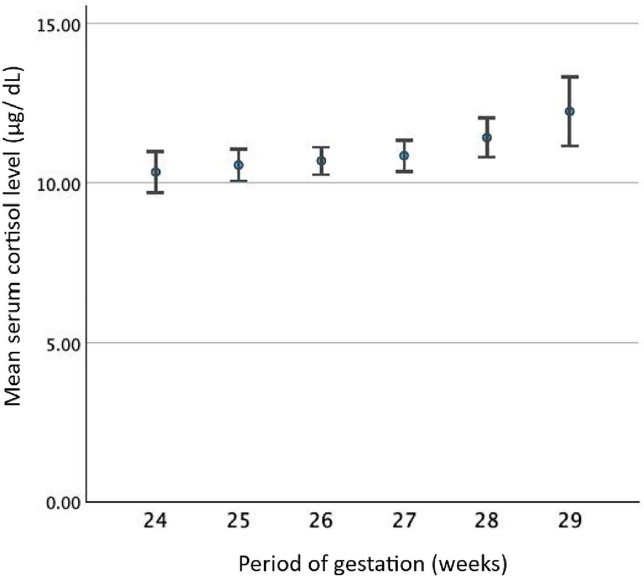

**Fig 2. Distribution of mean serum cortisol level (µg/dL) by the period of gestation (POG) in weeks.**

## Association of serum cortisol levels with sociodemographic and psychological factors

An EPDS score of more than 9 was not associated with serum cortisol ($p = 0.633$). The first-trimester EPDS scores of anhedonia (one-way ANOVA F(641,963)=1.01, $p = 0.444$), anxiety (one-way ANOVA F(641,963)=0.96, $p = 0.655$), or depression (due to low numbers with each score, participants with a score of 10 or more were amalgamated into one group) (one-way ANOVA F(641,963)=1.13, $p = 0.103$) in the first trimester did not predict mean serum cortisol in the second and third trimester (Table 2). Similarly, suicidal attempts ($p = 0.185$), suicidal ideation ($p = 0.572$), and attempts at deliberate self-harm ($p = 0.631$) were not associated with serum cortisol levels. The serum cortisol level was not associated with the EPDS score in the second trimester (one-way ANOVA F (599, 872) =0.96, $p = 0.663$).

## Independent predictors of serum cortisol level

The robust regression analysis showed that a history of miscarriages (ß = 0.081, 95%CI 0.020 to 0.142, $p = 0.010$) and POG at cortisol test (ß = 0.015, 95%CI 0.006 to 0.025, $p = 0.002$) were positively associated with serum cortisol levels, whereas BMI (ß = −0.004, 95%CI −0.009 to −0.001, $p = 0.044$) and gravidity (ß = −0.044, 95%CI −0.074 to −0.014, $p = 0.004$) were negatively associated (Table 3). These associations remained significant after adjusting for potential confounders.

## Pregnancy outcomes

Pregnancy outcomes were available for 990 participants (98.0%), including 985 live births (99.5%), 3 stillbirths (0.3%), and 2 intrauterine deaths ≥28 weeks of gestation (0.2%). Among live births, 4 early neonatal deaths (<7 days) and 5 infant deaths (<12 months) were reported. Preterm birth occurred in 208 cases (21.1%), including 10 very preterm (1.0%) and 3 extreme preterm deliveries (0.3%). Mean serum cortisol levels did not differ significantly between preterm (11.14 ± 3.74 µg/dL) and term births (10.87 ± 3.87 µg/dL; $p = 0.372$). Of 968 births with available weight data, 153 (15.8%) were low birth weight, 4 (0.4%) were very low birth weight, and none were extremely low birth weight. Cortisol levels were comparable between low birth weight (10.87 ± 3.53 µg/dL) and normal birth weight deliveries (10.94 ± 3.91 µg/dL; $p = 0.837$).

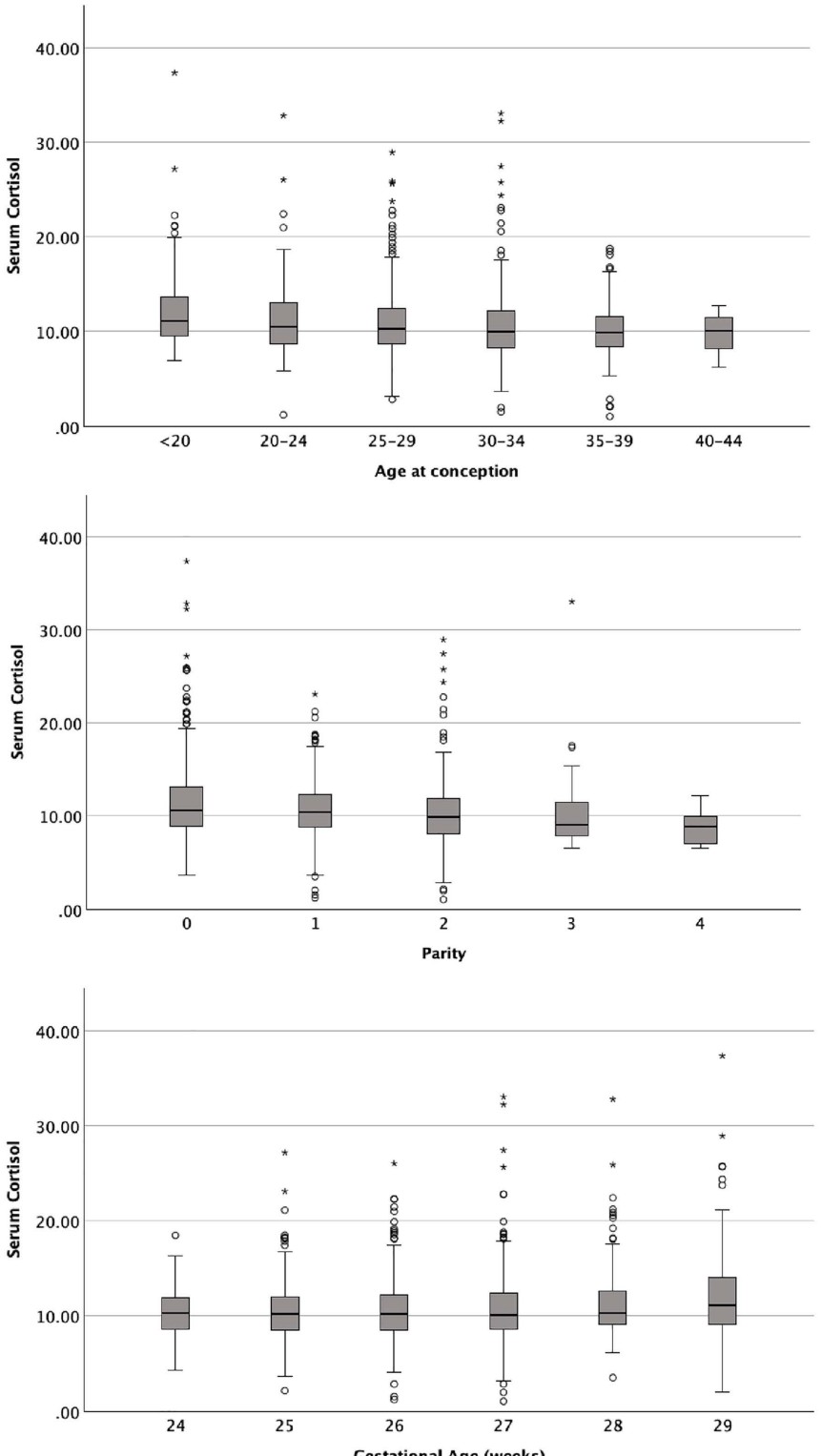

**Fig 3. Distribution of serum cortisol level by the age at conception in years (top), parity (middle), and gestational age in completed weeks (bottom).**

**Table 2. Sociodemographic factors and psychological factors affecting cortisol levels in the second and third trimesters in pregnant women from Anuradhapura, Sri Lanka (N = 1010).**

| | N | Percentage (%) | Cortisol level (µg/dL) | | | Significance |
|---|---|---|---|---|---|---|
| | | | Mean | Standard deviation | Median | |
| **Demographic factors** | | | | | | |
| **Marital status** (at recruitment) | | | | | | T = −1.01 |
| Married | 987 | 97.7 | 10.91 | 3.81 | 10.29 | p = 0.269 |
| Unmarried | 23 | 2.3 | 11.80 | 4.37 | 10.63 | |
| **Maternal ethnicity** | | | | | | F(2,1000)=0.41 |
| Sinhalese | 916 | 90.7 | 10.90 | 3.85 | 10.27 | p = 0.661 |
| Moor/ Malay | 84 | 8.3 | 11.24 | 3.71 | 10.53 | |
| Other | 10 | 1.0 | 11.48 | 2.35 | 11.53 | |
| **Maternal religion** | | | | | | F(2,1010)=0.83 |
| Buddhism | 908 | 89.9 | 10.91 | 3.86 | 10.29 | p = 0.438 |
| Islam | 89 | 8.9 | 11.26 | 3.66 | 10.18 | |
| Other | 13 | 1.3 | 9.89 | 1.61 | 9.45 | |
| **Highest grade completed in school (maternal)** | | | | | | F(2,1003)=0.07 |
| Grade 10 or less | 85 | 8.4 | 11.06 | 3.35 | 10.39 | p = 0.930 |
| Grade 11 | 515 | 51.0 | 10.96 | 3.81 | 10.34 | |
| Grade 12–13 | 403 | 39.9 | 10.90 | 3.96 | 10.18 | |
| **Paternal ethnicity** | | | | | | F(2,1007)=0.30 |
| Sinhalese | 913 | 90.4 | 10.90 | 3.86 | 10.26 | p = 0.741 |
| Moor/ Malay | 79 | 7.8 | 11.20 | 3.76 | 10.40 | |
| Other | 15 | 1.5 | 11.34 | 2.42 | 11.52 | |
| **Paternal religion** | | | | | | F(2,1008)=0.61 |
| Buddhism | 905 | 89.6 | 10.91 | 3.86 | 10.27 | p = 0.541 |
| Islam | 90 | 8.9 | 11.26 | 3.63 | 10.62 | |
| Other | 13 | 1.3 | 10.15 | 2.83 | 10.53 | |
| **Highest grade completed in school (paternal)** | | | | | | F(2,998)=1.05 |
| Grade 10 or less | 128 | 12.7 | 10.52 | 3.35 | 9.98 | p = 0.350 |
| Grade 11 | 567 | 56.1 | 11.06 | 4.08 | 10.31 | |
| Grade 12–13 | 301 | 30.0 | 10.92 | 3.55 | 10.47 | |
| **Psychological health** | | | | | | |
| **Edinburgh postpartum depression scale score quintile (first visit)** | | | | | | F(4,986)=1.245 |
| First | 219 | 22.20 | 11.02 | 3.44 | 10.53 | p = 0.290 |
| Second | 179 | 18.20 | 11.46 | 3.98 | 10.71 | |
| Third | 325 | 33.00 | 10.73 | 3.98 | 10.19 | |
| Fourth | 82 | 8.30 | 10.95 | 4.15 | 10.08 | |
| Fifth | 181 | 18.40 | 10.73 | 3.61 | 10.27 | |
| **Self-harm ideation during the past two weeks (first visit)** | | | | | | T = −0.12 |
| Yes often | 17 | 1.80 | 10.80 | 3.02 | 10.52 | p = 0.903 |
| No/ occasionally | 946 | 98.20 | 10.91 | 3.76 | 10.32 | |
| **Edinburgh postpartum depression scale score quintile (second visit)** | | | | | | F(3,672)=0.23 |
| First | 200 | 22.90 | 10.43 | 3.39 | 9.78 | p = 0.631 |
| Second | 180 | 20.60 | 11.06 | 4.04 | 10.25 | |
| Third | 205 | 23.50 | 11.08 | 3.82 | 10.43 | |
| Fourth | 143 | 16.40 | 11.39 | 3.62 | 10.59 | |
| Fifth | 144 | 16.50 | 10.8 | 4.42 | 10.34 | |

*(Continued)*

**Table 2.** (Continued)

| | N | Percentage (%) | Cortisol level (µg/dL) | | | Significance |
|---|---|---|---|---|---|---|
| | | | Mean | Standard deviation | Median | |
| **Self-harm ideation during the past two weeks (second visit)** | | | | | | T=0.91 |
| Yes often | 34 | 3.90 | 11.47 | 4.58 | 10.56 | p=0.367 |
| No/ occasionally | 849 | 96.10 | 10.88 | 3.7 | 10.25 | |
| **Deliberate self-harm attempts during the current pregnancy** | | | | | | |
| Yes | 13 | 1.40 | 11.44 | 5.23 | 10.10 | T=0.481 |
| No | 937 | 98.60 | 10.92 | 3.85 | 10.29 | p=0.631 |
| **Suicidal ideation during current pregnancy** | | | | | | T=0.57 |
| Yes | 11 | 1.20 | 11.58 | 5.31 | 10.10 | p=0.572 |
| No | 938 | 98.80 | 10.92 | 3.85 | 10.29 | |
| **Suicide attempts during current pregnancy** | | | | | | T=1.33 |
| Yes | 8 | 0.80 | 9.12 | 1.41 | 8.99 | p=0.185 |
| No | 940 | 99.20 | 10.94 | 3.88 | 10.3 | |
| **Lifetime perception of life as not worth living** | | | | | | T=1.06 |
| Yes often | 4 | 0.40 | 12.98 | 1.58 | 13.02 | p=0.287 |
| No/ occasionally | 946 | 99.60 | 10.92 | 3.87 | 10.27 | |
| **History of mental or physical abuse by an intimate partner** | | | | | | T=1.30 |
| Yes | 19 | 2.20% | 12.07 | 5.9 | 9.88 | p=0.194 |
| No | 850 | 97.80% | 10.91 | 3.8 | 10.3 | |
| **Perceived susceptibility to mental health issues** | | | | | | T=1.41 |
| Yes | 19 | 2.30% | 12.21 | 3.6 | 11.18 | p=0.157 |
| No | 819 | 97.70% | 10.94 | 3.89 | 10.28 | |

**Table 3. Robust Regression to determine independent predictors of serum cortisol level.**

| Variable | df | Coefficient | 95%CI | | Significance (p-value) |
|---|---|---|---|---|---|
| | | | lower | upper | |
| **Constant** | 822 | 2.164 | 1.787 | 2.540 | <0.001 |
| **Age** | 822 | −0.002 | −0.006 | 0.003 | 0.505 |
| **Education level** | 822 | −0.004 | −0.018 | 0.009 | 0.524 |
| **Marital status** | 822 | 0.036 | −0.101 | 0.173 | 0.605 |
| **Gravidity** | 822 | −0.044 | −0.074 | −0.014 | 0.004 |
| **History of miscarriages** | 822 | 0.081 | 0.020 | 0.142 | 0.010 |
| **Body-mass-index** | 822 | −0.004 | −0.009 | −0.001 | 0.044 |
| **Period of gestation at cortisol test** | 822 | 0.015 | 0.006 | 0.025 | 0.002 |

## Discussion

We report normative data for serum cortisol levels in pregnancy and examine the factors associated with maternal serum cortisol in a large population-based cohort from rural Sri Lanka for the first time in Sri Lanka. The mean serum cortisol level in the study sample was close to the lower limit, with none of the study participants exceeding the upper limit of the standard levels [32]. There was no significant difference between singleton and twin pregnancies. Serum cortisol levels in participants with 29-week POG were higher than in women with 24-week POG. The robust regression analysis showed that a history of miscarriages and POG at the cortisol test were positively associated with serum cortisol levels, whereas BMI and gravidity were negatively associated. We did not observe an association of serum cortisol with psychosocial factors.

## Normative levels

The current study includes all pregnant women enrolled in the maternal care program from the Anuradhapura district, Sri Lanka, validating the normative data presented. In this study, the mean serum cortisol level was low, and the upper limit of cortisol level was not exceeded in any participant. However, this result should be interpreted with caution as chronic stress may not result in elevated serum cortisol levels [34]. The mean cortisol levels reported from Southern Brazil in the first, second, and third trimesters were 28.5 µg/dl (95%CI: 26.1 to 31.2), 42.6 µg/dl (95%CI: 41.0 to 44.2), and 54.4 µg/dl (95%CI: 51.4 to 57.7), respectively with an increase of 2.0% (95%CI 1.01 to 1.02, $p < 0.001$) each week [35]. Normal levels reported in Melbourne, Australia, in the first, second, and third trimesters were $20.92 \pm 1.09$ µg/dL, $31.84 \pm 1.01$ µg/dL, and $37.81 \pm 1.49$ µg/dL, respectively. However, further prospective cohort studies should be conducted in different geographies and populations to corroborate these findings. The non-significant difference between singleton and twin pregnancies in the current study is probably attributable to the low number of twin pregnancies, which resulted in a large standard deviation despite the apparent difference in the mean. The increase in cortisol levels with POG follows the known pattern of increasing from the first to the third trimester and during labor [12,14,36].

## Maternal factors

We report significantly higher serum cortisol levels in primi, nulliparous, and teenage pregnant women and women with past miscarriages compared to their counterparts. Robust regression was used to account for the presence of outliers and deviations from the assumptions of standard linear regression. This approach enhanced the reliability of our findings, especially given the skewed nature of cortisol data and biological variability among participants. It showed similar results: a positive association between POG and a history of miscarriage, and an inverse relationship with BMI and gravidity. The negative association of gravidity with serum cortisol levels may reflect physiological or psychological adaptation in women with previous pregnancies [37]. It is possible that women who have experienced multiple pregnancies develop better coping mechanisms or reduced hypothalamic-pituitary-adrenal (HPA) axis reactivity over time. In contrast, the positive association between a history of miscarriages and cortisol levels even in the second and third trimesters may indicate increased stress or anxiety in subsequent pregnancies, which has not been adequately explored. The negative association of gravidity with serum cortisol levels may reflect physiological or psychological adaptation in women with previous pregnancies [37]. It is possible that women who have experienced multiple pregnancies develop better coping mechanisms or reduced hypothalamic-pituitary-adrenal (HPA) axis reactivity over time. In contrast, the positive association between a history of miscarriages and cortisol levels even in the second and third trimesters may indicate increased stress or anxiety in subsequent pregnancies, which has not been adequately explored. Similar to the current study, there was a significant difference between the mean serum cortisol levels of underweight, normal-weight, overweight, and obese individuals in other studies [38]. Several studies have shown that women with pre-pregnancy obesity have a lower level of salivary [39] and serum cortisol compared to lean women without an increase in urinary glucocorticoid clearance, suggestive of decreased maternal hypothalamic-pituitary axis activity [40].

Known stressors in early pregnancy, such as marital status and contraceptive failure, were not significantly associated with serum cortisol levels in this study. However, contrary results similar to the current study were reported from a prospective cohort study from Bangalore, where a significant difference in cortisol levels was found for maternal age and gravidity, but no other pregnancy-related stressors such as history of abortion, social support, and spousal alcohol consumption or smoking [15].

## Serum cortisol and psychosocial factors

Current study results did not show an association between the serum cortisol level in 24–29 weeks of POG with the EPDS score or other psychosocial factors. However, this may be attributable to the extensive supportive program, which

included targeted interventions such as counseling, health education, financial and social support, and psychiatric referral as appropriate, conducted for the participants with high EPDS or psychological risk factors in the first-trimester screening [23]. Previous studies have shown mixed results on the association between subjective measures of depression or stress and serum cortisol levels [41,42]. However, current evidence favors maternal cortisol levels over subjective measures as a predictor of offspring outcomes [11]. Therefore, it is recommended to assess maternal cortisol in future studies and explore the public health implications of high maternal serum cortisol.

## Strengths and limitations

This is a prospective cohort study with a large sample size with only 3 other studies having a larger sample size, which were mainly from the developed countries. Furthermore, this is a population-based cohort study including all pregnant women residing in the Anuradhapura district, North Central Province, Sri Lanka, and registered with the national maternal care program from 1st July to 30th September 2019 and with a POG of less than 13 weeks. The national maternal care program has 95% antenatal coverage. Therefore, this study sample includes a representative population. A structured questionnaire administered by trained medical care personnel was used to evaluate psychological stressors. Furthermore, physical measurements were conducted by trained medical personnel. The fasting early-morning serum cortisol levels were evaluated in all participants after an adequate test by a trained phlebotomist and evaluated on the same day at an accredited laboratory. The current study did not evaluate hair or salivary cortisol levels for corroboration with serum cortisol levels due to budgetary concerns and the lack of accredited laboratory facilities.

## Conclusion

The mean serum cortisol level in all pregnant women was 10.93(±3.83) µg/dL, with none of the participants exceeding the upper limit and with no significant difference between singleton and twin pregnancies. Serum cortisol levels gradually increased with the POG, with a significant difference observed between POG 24 and POG 29 weeks. A history of miscarriages and POG at cortisol test were positively associated with serum cortisol levels, whereas BMI was negatively associated. A high EPDS score or other psychological stressors were not associated with the serum cortisol level.

## Acknowledgments

The author Shashanka Rajapakse expresses their appreciation for the support from the International Cooperation & Education Program (NCCRI·NCCI 52210–52211, 2024) of the National Cancer Center, Korea.

## Author contributions

**Conceptualization:** Ishani Menike, Shashanka Rajapakse, Thilini Agampodi, Suneth Buddhika Agampodi.

**Data curation:** Ishani Menike, Shashanka Rajapakse, Gayani Amarasinghe, Suneth Buddhika Agampodi.

**Formal analysis:** Ishani Menike, Shashanka Rajapakse, Janith Warnasekara, Suneth Buddhika Agampodi.

**Funding acquisition:** Thilini Agampodi, Suneth Buddhika Agampodi.

**Investigation:** Ishani Menike, Gayani Amarasinghe, Nuwan Darshana Wickramasinghe, Thilini Agampodi, Suneth Buddhika Agampodi.

**Methodology:** Ishani Menike, Shashanka Rajapakse, Gayani Amarasinghe, Nuwan Darshana Wickramasinghe, Thilini Agampodi, Suneth Buddhika Agampodi.

**Project administration:** Nuwan Darshana Wickramasinghe, Thilini Agampodi, Suneth Buddhika Agampodi.

**Supervision:** Suneth Buddhika Agampodi.

**Visualization:** Ishani Menike, Shashanka Rajapakse, Suneth Buddhika Agampodi.

Writing – original draft: Ishani Menike, Shashanka Rajapakse.

Writing – review & editing: Gayani Amarasinghe, Janith Warnasekara, Nuwan Darshana Wickramasinghe, Thilini Agampodi, Suneth Buddhika Agampodi.

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
