## [Decision Letter · Decision Letter 0]

24 Jun 2025

Dear Dr. Agampodi,

Thank you for submitting your manuscript to PLOS ONE. After careful consideration, we feel that it has merit but does not fully meet PLOS ONE’s publication criteria as it currently stands. Therefore, we invite you to submit a revised version of the manuscript that addresses the points raised during the review process.

**ACADEMIC EDITOR:**

The authors are to be commended for the successful submission of this manuscript, which addresses an important area of research with potential implications for maternal and fetal health. The use of a population-based prospective cohort design is a notable strength.<o:p></o:p>

However, there is some ambiguity regarding the timing of participant recruitment. While it is stated under the "Procedure" section (line 153) that recruitment occurred during the first trimester, it is also mentioned (lines 146–147) that serum cortisol levels were available only for 1,290 participants who attended follow-up visits in the second or third trimester. Although the authors refer to a previously published methodology, it would greatly improve the clarity of the current manuscript to explicitly state when participants included in this analysis were initially recruited. Clear delineation of the recruitment timeline would aid in understanding the study design and cohort selection.<o:p></o:p>

We look forward to receiving your revised manuscript.

Kind regards,

Surangi Jayakody, MBBS, MSc, MD

Academic Editor

PLOS ONE

Journal Requirements:

“The original cohort study was supported by the Accelerating Higher Education Expansion and Development (AHEAD) Operation of the Ministry of Higher Education, Sri Lanka funded by the World Bank [grant number DOR STEM HEMS [6026-LK/8743-LK]]. The funding agency has no role in the design of the study and collection, analysis, and interpretation of data and in writing the manuscript.”

Reviewers' comments:

Reviewer's Responses to Questions

**Comments to the Author**

1. Is the manuscript technically sound, and do the data support the conclusions?

Reviewer #1: Partly

Reviewer #2: Yes

2. Has the statistical analysis been performed appropriately and rigorously?

Reviewer #1: Yes

Reviewer #2: I Don't Know

3. Have the authors made all data underlying the findings in their manuscript fully available?

Reviewer #1: Yes

Reviewer #2: Yes

4. Is the manuscript presented in an intelligible fashion and written in standard English?

Reviewer #1: Yes

Reviewer #2: Yes

Reviewer #1: I read the presented manuscript with great interest. I wish to commend the authors in presenting a large data set from a LMIC set up which is a difficult task to accomplish. The study presents descriptive data of maternal serum cortisol levels in a large cohort of pregnant women in the second trimester.

I have only a few comments to make.

1. Please highlight the fact that ALL women in the cohort had their cortisol levels with-in the accepted reference range for the gestation.

2. I agree with the authors that the relationship with EPDS score and maternal serum cortisol levels may not have shown a correlation due to the chronology of the measurements. Please make a clear statement of when the EPDS score was obtained and when the serum cortisol level was measured- what was the time difference between the two measurements.

3. If the authors can present pregnancy outcome data, please check for fetal outcomes in relationship with second trimester cortisol levels - eg, previous authors have presented a relationship with high normal (>17) second trimester cortisol levels with adverse fetal outcomes. (The data will be an useful addition for a systematic review/ meta analysis)

Reviewer #2: Thank you for conducting this research and comprehensive article . This gives eye open for pregnancy and its related factors in developing country. I thought increase in cortisol levels in psychological stresses but it was not. Very interesting article . Best wishes for future endeavours

**Do you want your identity to be public for this peer review?** For information about this choice, including consent withdrawal, please see our Privacy Policy

Reviewer #1: No

Reviewer #2: **Yes: ** Dr. Chamara Wijemunige- Speciality Registrar in Obs & Gyne

---

## [Author Response · Author response to Decision Letter 1]

22 Jul 2025

Response letter

The authors are highly appreciative and grateful for the excellent review process. We are delighted with the constructive, encouraging, and insightful comments of the editor and the reviewers. We strongly believe that the comments of the editor and the reviewers greatly enhanced the quality and the scientific impact of our manuscript. Furthermore, we are thankful for the comments to improve the comprehensibility and the understandability of our manuscript. We addressed all the comments and enhanced our manuscript as suggested.

Please see the detailed responce to all comments in the attched responce letter.

---

## [Editor Report · Decision Letter 1]

6 Aug 2025

Normative serum cortisol levels in second and third trimesters and their associated factors: A prospective cohort study from Sri Lanka

PONE-D-25-16021R1

Dear Dr. Agampodi,

We’re pleased to inform you that your manuscript has been judged scientifically suitable for publication and will be formally accepted for publication once it meets all outstanding technical requirements.

Kind regards,

Surangi Jayakody, MBBS, MSc, MD

Academic Editor

PLOS ONE
---

## [Editor Report · Acceptance letter]

PONE-D-25-16021R1

PLOS ONE

Dear Dr. Agampodi,

I'm pleased to inform you that your manuscript has been deemed suitable for publication in PLOS ONE. Congratulations! Your manuscript is now being handed over to our production team.

Kind regards,

on behalf of

Dr Surangi Jayakody

Academic Editor

PLOS ONE